# A Biotechnological Approach for the Production of Pharmaceutically Active Human Interferon-α from *Raphanus sativus* L. Plants

**DOI:** 10.3390/bioengineering9080381

**Published:** 2022-08-10

**Authors:** Rashad Kebeish, Emad Hamdy, Omar Al-Zoubi, Talaat Habeeb, Raha Osailan, Yassin El-Ayouty

**Affiliations:** 1Botany and Microbiology Department, Faculty of Science, Zagazig University, Zagazig 44519, Egypt; 2Biology Department, Faculty of Science Yanbu, Taibah University, Yanbu El-Bahr 46423, Saudi Arabia

**Keywords:** *Raphanus sativus* L., recombinant IFN-α2a, apoptosis, antiviral and antitumor activity

## Abstract

Human interferon (IFN) is a type of cytokine that regulates the immune system’s response to viral and bacterial infections. Recombinant IFN-α has been approved for use in the treatment of a variety of viral infections as well as an anticancer medication for various forms of leukemia. The objective of the current study is to produce a functionally active recombinant human IFN-α2a from transgenic *Raphanus sativus* L. plants. Therefore, a binary plant expression construct containing the IFN-α2a gene coding sequence, under the regulation of the cauliflower mosaic virus 35SS promoter, was established. *Agrobacterium*-mediated floral dip transformation was used to introduce the IFN-α2a expression cassette into the nuclear genome of red and white rooted *Raphanus sativus* L. plants. From each genotype, three independent transgenic lines were established. The anticancer and antiviral activities of the partially purified recombinant IFN-α2a proteins were examined. The isolated IFN-α2a has been demonstrated to inhibit the spread of the Vesicular Stomatitis Virus (VSV). In addition, cytotoxicity and cell apoptosis assays against Hep-G2 cells (Human Hepatocellular Carcinoma) show the efficacy of the generated IFN-α2a as an anticancer agent. In comparison to bacterial, yeast, and animal cell culture systems, the overall observed results demonstrated the efficacy of using *Raphanus sativus* L. plants as a safe, cost-effective, and easy-to-use expression system for generating active human IFN-α2a.

## 1. Introduction

Some human disorders are caused by protein deficiency and poor performance [1]. Protein medicines are expected to be an armory against illnesses and have enormous commercial value [2]. Many advantages of molecular farming have been discovered in recent years, particularly in terms of cost, practicality, and safety. Transgenic plants have been used in the development of vaccines and therapeutic proteins in the biopharmaceutical field [2,3]. Plants such as rice [4], aloe [5], tobacco [6], and *Raphanus sativus* can be used to make edible vaccines for oral delivery and immunization. Currently, new research is focusing on figuring out how to make them inexpensive, easily produced, and fully functional. Previously, routinely produced proteins were taken from natural sources for use in research, medicine, and industry [7]. The protein generated using this approach rarely met the requirements, and it also posed several risks and challenges in isolation [2]. Sorensen and Mortensen, (2005) [8] believe that biotechnological technologies are the only way to address the demand for pure, soluble, and functional proteins. To meet the capital required as well as the production scale-up and efficacy, numerous expression systems were developed, including bacterial, yeast, animal cell lines, and plants [9]. As a result, deciding on an expression system necessitates a cost breakdown in terms of design, procedure, and other economic factors. Because its genome has been fully sequenced and the organism is easy to handle, grows quickly, and requires an inexpensive, easy-to-prepare medium for growth, *E. coli* has been the “factory” of choice for production of numerous recombinant proteins [10]. Since glycosylation and post-translational modifications do not occur in *E. coli*, many eukaryotic proteins generated are in a nonfunctional, incomplete state. For protein manufacturing, scientists have resorted to eukaryotic yeast and mammalian expression systems [11]. For the synthesis of foreign proteins, eukaryotic yeast is an excellent host. It is free of the endotoxin problem that bacteria have. Because yeast produces proteins that are appropriately folded and secreted into the media, the yeast system has been used to make many beneficial proteins, including human serum albumin, tetanus toxin fragment, and lysozymes [12]. Current manufacturing processes rely on fermentation technologies, which necessitate sterile factories and complex purification techniques. These technologies are costly, and the building and certification of manufacturing facilities can take up to 4–5 years [13]. Transgenic organisms (crops/animals) are used in modern biotechnology to obtain vast amounts of complex proteins at a low cost. Plant expression systems provide cheap cultivation costs, high biomass production, a short period from gene to protein, low capital and operating expenses, and excellent scalability with better protein yields [14,15]. Transient expression and steady transformation are two main strategies for producing recombinant proteins in plants. They are unlikely to contaminate human pathogens or endotoxins. Although many recombinant proteins have been successfully synthesized in plant systems, only a few have been approved for human use, and others are still being evaluated in clinical trials [16].

Interferons are a family of cytokines that play important roles in antiviral and antibacterial defenses, immune system activation, and cell growth control [17]. IFN is a protein that is mostly utilized to treat viral infections and malignant neoplasms. When compared to other medications, IFN has several advantages, including a defined duration of treatment, absence of drug-resistant variations, and long-term efficacy [18]. In 2014, the global interferon trade reached over four billion dollars [19]. Because of the high cost and great demand for interferon, researchers are focusing their efforts on developing cost-effective and safe protein production technologies [17]. The objective of the current study is to generate and assess the biological activity of a recombinant human interferon-α2a produced from *Raphanus sativus* L. plants. Therefore, the human interferon-α2a (IFN-α2a) coding sequence was genetically cloned into a binary expression construct:pTRA-PT vector. This plant expression construct contains the PAT gene (phosphinothricin acetyltransferase) that is able to resist phosphinothricin herbicide, allowing the selection of transgenic plants upon the foliar spray of the herbicide [20]. The PAT enzyme is innocuous with high specificity to degrade phosphinothricin. It does not have any characteristics associated with allergens or food toxins and the enzyme is degraded by intestinal and gastric fluids [21]. The construct was transferred to *Raphanus sativus* L. plants via the *Agrobacterium*-mediated floral dip transformation method. A single *Raphanus sativus* (red and white) plant is capable of generating large amounts of seed and a large amount of shoot biomass per year [22]. As a bioreactor, *Raphanus sativus* L. plants could be considered to be less expensive than the frequently used *E. coli* fermentation systems that lack glycosylation and the post-translational modifications needed for fully active human interferon-α [23]. Three lines from each *Raphanus sativus* L. transgenic plant have been generated. Protein extracts from each genotype were subjected to column purification using DEAE-Sepharose and Sephadex G-50 in order to obtain enriched IFN-α2a protein fractions for evaluating the biological activity of the recombinant IFN-α2a as an antiviral and anticancer agent.

## 2. Materials and Methods

### 2.1. Plant Material

In this study, white and red radish (*Raphanus sativus* L.) plants were used for *Agrobacterium*-mediated floral dip transformation. Seeds of both radish varieties were obtained from the Seed and Plant Improvement Center of the Egyptian Ministry of Agriculture, Giza, Egypt.

### 2.2. Plasmid Construct

PCR primers (5′-TGATCCATGGCCTTGACCTTTGCTTTACTG-3′ as the forward primer and 5′-GTGCTCTAGATCATTCCTTACTTCTTAATC-3′ as the reverse primer) were used to amplify human IFN-α 2a (GenBank: JN848522.1). The primers were created with *Nco* I and *Xba* I site extensions, respectively. The primer annealing temperature was 56 °C and the MgCl_2_ concentration used in the PCR buffer was 2.5 mM. Because IFN-2a genomic DNA sequence contains no introns [24], DNA extracted from human blood was used as a PCR template. IFN-α2a PCR product was column purified (Qiagen, Darmstadt, Germany), then cut with *Xba* I and *Nco* I restriction enzymes before being ligated into the pTRA-PT vector (gi13508478). IFN-α2a was then confirmed by sequencing. The scaffold attachment regions (SAR) of tobacco RB7 gene (gi3522871), 5′ UTR of tobacco leader peptide (TL), and 3′ UTR of CaMV 35S (pA35S) flanked the expression cassette of the IFN-α2a gene coding sequence. The expression of the IFN-α2a gene is influenced by the CaMV 35S promoter. The selection of transgenic lines of white and red *Raphanus sativus* L. was performed by spraying the foliar parts of 3 weeks old plantlets with BASTA (phosphinothricin 25 µg/mL) 6 times at 3-day intervals based on the presence of the PAT (phosphinothricin acetyltransferase) expression cassette. The structure of the used binary expression vector was previously described [20] and the structure of the IFNα2a expression cassette is illustrated in Figure 1.

### 2.3. Transformation and Selection of Transgenic Raphanus sativus L.

The *Agrobacterium tumefaciens* (GV3101)-mediated floral dip transformation protocol as described by Curtis and Nam (2001) [25] was applied to transform wild-type white and red *Raphanus sativus* L. plants with pTRA-PT-IFN-α2a construct, see Figure 2A. Three-week-old T_1_ plantlets were subjected to BASTA (phosphinothricin 25 µg/mL) herbicide application six times at three days intervals, see Figure 2B. Phosphinothricin is a broad-spectrum herbicide that inhibits glutamine synthetase enzyme in plants. This herbicide represents a low risk of environmental contamination due to its fast degradation in soil and water. T_1_ plants used for molecular, biochemical, antiviral, and antitumor assays were grown at ambient conditions (14 h light and 10 h dark) in a greenhouse, see Figure 2C. To confirm the presence of IFN-α2a gene in the selected T_1_ plants of white and red *Raphanus sativus* L., genomic DNA was isolated from transgenic lines. The isolated DNA was tested by PCR using vector-specific primers located in the promoter and terminator regions flanking the IFN-α2a gene (forward primer: 5′-GAC CCT TCC TCT ATA TAA GG-3′ and reverse primer: 5′-CAC ACA TTA TTC TGG AGA AA-3′), see Figure 3.

### 2.4. Real-Time RT-PCR

Quantitative real-time PCR was performed as described previously [26]. The BCP (1-bromo-3-chloropropane) protocol [27] was applied to extract RNA from transgenic *Raphanus sativus* L. leaves. The isolated RNA was subjected to DNase-1 digestion before proceeding to cDNA synthesis. First-strand cDNA synthesis was implemented as previously described [28]. RT-PCR was performed using the StepOne qRT-PCR system (Applied Biosystems, Waltham, MA, USA) following the manufacturer’s instructions in the presence of SYBR Green (SYBR1 GreenERTM qPCR SuperMixes; Karlsruhe, Germany). Primers were purchased from Intron Biotechnology Inc. (Kyungki-Do, Korea). 5′-CTG AAA CCA TCC CTG TCC TC 3′ and 5′-TCT AGG AGG GTC TCA TCC CA-3′ primers were used to detect IFN-α2a transcripts. For the detection of the housekeeping gene, *ACTIN2* transcripts was performed using 5′-GGTAACATTGTGCTCAGTGGTGG-3′ and 5′-GGTGCAACGACCTTAATCTTCAT-3′ primers. A final concentration of 200 nM primer was applied to the reaction mixture. The amplification program used for both *ACTIN2* and IFN-α2a was 10 min of initial denaturation at 95 °C, followed by 40 cycles each of 15 s denaturation at 95 °C and 1 min of annealing and extension at 60 °C.

### 2.5. IFN-α2a Protein Extraction and Partial Purification

100 mg frozen green leaves collected from 5-week-old WT and transgenic *Raphanus sativus* L. plants were homogenized in liquid nitrogen. 500 µLof protein extraction buffer (50 mM sodium phosphate buffer (pH 7.4) supplemented with 0.1% Triton X-100 and 10 mM EDTA) was added followed by centrifugation (30,000× *g*/15 min/4 °C). The supernatant was collected and the centrifugation step was repeated. Partial purification of IFN-α2a protein samples was performed using DEA-Sepharose followed by Sephadex G-50 (Sigma Aldrich, Taufkirchen, Germany). Protein concentration was evaluated based on the Bradford method [29]. Total protein extracts and enriched IFN-α2a protein fractions were kept in freezers at −20 °C and used for further analyses.

### 2.6. Enzyme-Linked Immunosorbent Assay (ELISA) for Detection of IFN-α2a Protein

Enzyme-linked Immunosorbent Assay (ELISA) was applied for the detection of recombinant IFN-α2a protein using an IFN alpha Human ELISA Kit (ThermoFisher Scientific, Altrincham, UK). Detection and quantification of IFN-α2a in protein extracts were carried out according to the supplier’s protocol. The titer of interferon in test samples was calculated by plotting the optical densities (450 nm) and fitting the standard curve with a 6-parameter fit. Peg-IFN (Pegasys^®^; Hoffmann-LaRoche, Basel, Switzerland), the commercially available IFN-α, was used as a positive control in this assay. To eliminate the background, the blank sample values were subtracted from the standards and the test samples.

### 2.7. Antiviral Activity of the Recombinant IFN-α2a

Vesicular Stomatitis Virus (VSV) Indiana strain-ATCC^®^ VR-158 and its host cells (Vero cells, derived from the kidney of a normal, adult African green monkey) were generously provided by the Head of R&D sector VACSERA-Egypt, Prof. Dr. Aly Fahmy. Vero cells were allowed to grow as a monolayer in RPMI medium containing L-glutamine, sodium pyruvate, sodium bicarbonate, non-essential amino acids, 10% fetal bovine serum (FBS), and antibiotics (10 μg/mL streptomycin and 100 μg/mL penicillin) at 37 °C in a 5% CO_2_ humidified environment until the formation of confluent cell monolayers. VSV was prepared on Vero cells, aliquoted, and stored at −80 °C. Infectivity titer of VSV stock was determined as previously described [30]. The virus stock titer was 10^7^ TCID50/mL. 0.02 mL/well virus dilution was dispensed to pre-cultured Vero cells. In MEM (Sigma Aldrich, Taufkirchen, Germany) supplemented with 5% FCS (GEPCO, Miami, FL, USA), non-cytotoxic concentrations of recombinant IFN-α2a protein fractions were 10-fold serially diluted. To each concentration, equal volumes (~20μL) of virus at an infective titer of 10^2^ TCID50/mLwere added and incubated at 37 °C for 3h. To ensure an even distribution of virus on the cell surface, virus inocula were shaken at 15 min intervals. Culture plates were examined daily using an inverted microscope (Willovert, Helmut Hund GmbH, Wetzlar, Germany) until more than 90% cytopathic effect (CPE) was detected. As described by Lansky and Newman (2007) [31], 50% endpoint-induced CPE was determined. The antiviral activity of *Raphanus sativus* L. recombinant IFN-α2a protein was assessed both directly and indirectly based on its inhibitory effect against the viral cytopathic effect. Extracts from wild-type white and red *Raphanus sativus* L. and peg-IFN were used as negative and positive controls, respectively.

### 2.8. In Vitro CPE Assay of the Recombinant IFN-α2a on Hep-G2-Cells

Hep-G2 ATCC^®^ HB-8065 (human hepatocellular carcinoma cell line) was used to evaluate the antitumor properties of the recombinant IFN-α2a isolated from transgenic *Raphanus sativus* L. lines. This assay was performed at VACSERA, Cairo, Egypt. The assay is based on the mitochondrial-dependent reduction of the yellow MTT (3-(4,5-dimethylthiazol-2-yl)-2,5-diphenyl tetrazolium bromide) to purple formazan as previously described [32]. In brief, Hep-G2 cells were seeded with 100 µL of culture medium per well in 96-well plates. Cells were cultured alone (control) or with dilution series (final concentration of 10 µg/mL, 25 µg/mL, 50 µg/mL, 75 µg/mL, 100 µg/mL, 200 µg/mL, 300 µg/mL, 500 µg/mL, 750 µg/mL, and 1000 µg/mL) of the recombinant IFN-α2a protein extracts in DMSO. Wild-type *Raphanus sativus* L. protein extracts were used as negative controls. The experiment was performed in quadruplicate. The cells were grown in a 5% CO_2_ humidified incubator at 37 °C overnight. After discarding the growth medium, 100 μL of MTT working solution (0.4 mg/mL of PBS) was added and incubated for 3 h.The supernatant was then removed and 100 μL DMSO was added to each sample and absorbance at 550 nm was measured for each sample. The absorbance was calculated as a percentage of the control value. The change in viability percentage was calculated based on the formula (reading of extract/reading of negative control) *×* 100 CPE was also measured for normal Vero cells and WISH cells in order to determine the IC_50_ value of the recombinant IFN-α2a protein isolated from *Raphanus sativus* L. transgenic lines. IC_50_ values were calculated based on sigmoid concentration–response curve fitting models using Graph Pad Prism computer software (International scientific community, San Diego, CA, USA).

### 2.9. Hep-G2-Cell Apoptosis Assay

In 25 cm^2^ surface area cell culture flasks, Hep-G2 cells were pre-cultured and treated for 24h with the IC_50_ concentrations of the tested samples of recombinant IFN-α2a isolated from *Raphanus sativus* L. transgenic lines in RPMI-1640 and DMEM-media (Sigma Aldrich, Taufkirchen, Germany). For cell cycle analysis, Hep-G2 cells were collected and fixed gently with 70% (*v*/*v*) ethanol in FBS (Sigma Aldrich, Taufkirchen, Germany) overnight at 4 °C and then resuspended in FBS containing 40 μg/mLpenicillin, 0.1% (*v*/*v*) Triton X-100, and 0.1 mg/mL RNase in a dark room. Cells were incubated at 37 °C for 30 min then analyzed using a flowcytometer (Becton Dickinson, San Jose, CA, USA) equipped with an argon ion laser at a wavelength of 488 nm. Multicycle Software (Phoenix Flow Systems, San Diego, CA, USA) was used to evaluate the data, as previously reported [33].

### 2.10. Statistical Analysis

At least three independent tests were carried out for each experiment. Data were expressed as average values ± standard error (SE) and statistically analyzed using one-way analysis of variance (ANOVA). All statistical analyses were performed using the SPSS-11 program and Excel software (Microsoft Corporation, Redmond, WA, USA).

## 3. Results and Discussion

### 3.1. Generation and Selection of Transgenic Raphanus sativus L. plants

Plant molecular farming is a growing technique used to produce recombinant biopharmaceuticals, secondary metabolites, and other industrial proteins all over the world. The majority of this technology is based on introducing a gene or gene clusters into plants and/or plant cell cultures via molecular transformation. In 1986 and 1989, transgenic plants produced human growth hormone, the first recombinant pharmaceutical protein, and the first recombinant antibody, respectively [34,35]. Green plants and/or plant cell culture-based systems can be employed as large-scale biofactories for the manufacture of recombinant proteins [17,36,37]. *Raphanus sativus* L., an edible plant from the Cruciferae family, is an annual vegetable. It is grown mostly for its roots, which are consumed in salads by the majority of people. It can be eaten raw, cooked, or preserved by storage, canning, or drying. Despite this, radish is a valuable source of medicinal compounds [38]. *Raphanus sativus* L. is a promising option for building an efficient molecular farming system for the generation of human IFN-α2a because of its favorable growth properties, ease of genetic transformation [25], economical culture, and increased biosafety issues. As an eukaryotic expression system, *Raphanus sativus* L. plants can perform glycosylation as well as post-translational modification needed for fully active proteins, including human interferon-α [23].

*Raphanus sativus* L., a white and red rooted radish, was chosen as a test plant in this work with the goal of finding an alternate expression system for the generation of active human IFN-α2a. PCR was used to amplify the human IFN-α2a gene coding sequence (GenBank: JN848522.1). Because the genomic IFN-α2a on chromosome 9 lacks intron sequences, human genomic DNA isolated from blood was employed directly as a template [24]. The PCR product of IFN-α2a was amplified and cloned into the binary expression vector pTRA-PT (gi13508478). In green plants and/or microalgae, CaMV 35S promoter has been demonstrated to enhance high gene expression [20,26,39]. CaMV 35S promoter was therefore employed in the current study to regulate IFN-α2a gene expression in radish plants. The 5′ UTR of tobacco leader peptide (TL) and the 3′ UTR of CaMV 35S (pA35S) surround the IFN-2a expression cassette in the binary expression vector pTRA-PT (Figure 1).

*Agrobacterium tumefaciens* (GV3101) was transformed with pTRA-PT-IFN-α2a construct which was then utilized to transform wild-type white and red rooted radish plants using the floral dip transformation methodology [25] (Figure 2A). For the selection of transgenic lines, three-week-old T_1_ plantlets were sprayed with BASTA herbicide (phosphinothricin 25 µg/mL) six times at three days intervals, see Figure 2B,C. PCR analysis of genomic DNA obtained from radish plants revealed the presence of the IFN-α2a gene in the selected T_1_ transgenic radish plants. The result of this PCR screening is shown in Figure 3. Twelve white and six red *Raphanus sativus* L. were tested. Six plant samples from white radish (Lane 6–12 in Figure 3) and four samples of red radish (Lane 13, 14, 16, and 17 in Figure 3) confirm the existence of the IFN-α2a gene. Five transgenic white radishes and four transgenic red radishes were chosen for additional IFN-α2a gene expression analysis.

### 3.2. Analysis of IFN-α2a Expression in Transgenic White and Red Raphanus sativus L. plants

#### 3.2.1. RT-PCR Analysis

Quantitative RT-PCR was used to compare the accumulation of IFN-α2a mRNA transcripts in five transgenic white radish lines (named IFN-wRs-1-5) and four transgenic red radish lines (named IFN-rRs-1-4) to the housekeeping gene, Actin2 transcripts (Figure 4A,B respectively and Appendix A). Variable IFN-α2a expression levels in both white and red transgenic radish were observed. It is noted that transgenic white radish lines, IFN-wRs-1 & IFN-wRs-2, showed a relatively higher expression level of IFN-α2a compared to transgenic red radish lines. Similar expression levels of the IFN-α2a gene were observed for IFN-wRs-3, 4, 5, and all the transgenic red radish lines under the assay conditions. The higher IFN-α2a expressing transgenic lines from both radish types were chosen for further analyses. A similar expression pattern of human interferon was reported for IFN transgene in green plants [17], and/or microalgae [39]. However, for both bio-systems to elicit large amounts of human interferon production, codon use adaption of the introduced transgene is required.

#### 3.2.2. Enzyme-Linked Immunosorbent Assay (ELISA) Detection of Human IFN-α2a and SDS-PAGE Analysis

The presence of the IFN-α2a transgene in transformed *Raphanus sativus* L. plants was validated at the protein level by utilizing an Enzyme-linked Immunosorbent Assay (ELISA) kit to detect IFN-α2a protein (ThermoFisher Scientific, Altrincham, UK). Peg-IFN-α (Pegasys^®^; Hoffmann-LaRoche, Basel, Switzerland) was used in this experiment as a positive control. The kit includes a double-antibody sandwich that is used to detect the presence of human interferon (IFN-α) in test samples. Total protein extracts isolated from transgenic white and red radish lines, as well as wild-type control samples, were tested to detect IFN-α2a in the transgenic plant protein extracts using a commercially available IFN-α protein standard sample (Figure 5).

OD_450_ values recorded for IFN- α2a proteins in test samples by ELISA assay werefound to be highly correlated with the accumulation of IFN-α2a RNA transcripts (Figure 4A,B and Figure 5). Human IFN-α2a expressed in transgenic *Chlamydomonas reinhardtii* [39] and *E. coli* [40,41] was previously detected using an ELISA assay for human IFN-α2a.

Isolated total protein extracts from IFN-wRs-2 and IFN-rRs-2 transgenic lines were partially purified using DEA-Sepharose and then Sephadex G-50 (Sigma Aldrich, Taufkirchen, Germany). In these purification procedures, total protein extracts isolated from wild-type white and red radish plants were used as controls. The partially purified protein samples were subjected to SDS-PAGE analysis. A protein band in the size range of 19.5 kDa corresponding to IFN-α2a protein was observed for protein extracts isolated from transgenic IFN-wRs-2 and IFN-rRs-2 lines since this protein band was totally absent in wild-type control samples, see Appendix A. Moreover, the partially purified and total protein extracts of IFN-α2a were also subjected to UV-spectral scanning and compared to Peg-IFN control to confirm the existence of IFN-α2a protein in transgenic plant protein extracts (see Appendix A). Overall, the results of the qRT-PCR, IFN-α2a ELISA detection assays, SDS-PAGE assay of partially purified IFN-α2a protein isolated from transgenic plants, and UV-spectral scanning assays show that the IFN-α2a gene is efficiently expressed in transgenic *Raphanus sativus* L. lines.

The most commonly used bio-systems for the manufacturing of biopharmaceutical medications are animal cell cultures and microorganisms. Some of the recombinant human proteins created in microbes, including human IFN-α [23], do not have proper activity because microorganisms lack glycosylation and post-translational modifications of proteins. Transgenic animal, mammalian, insect cell cultures, and microbial fermentation systems are probably unsafe and not recommended for the production of biopharmaceuticals [2,42]. As a result, low-cost, simple, and effective technologies are required to meet the demand for large-scale and safe recombinant protein production for human applications. Plant-based systems are particularly cost-effective for producing recombinant proteins, costing around 1% of the cost of mammalian cells and 2–10% of the cost of microbial fermentation. Plants are therefore good candidates for supplying medicinal compounds [42,43]. Radish is known as one of the richest sources of medicinal compounds [38]. Other advantages, including ease of transformation, stability of gene expression, and its safety aspects concerning human consumption, all support the choice of radish as a bio-system for the production of IFN-α2a protein.

Protein extracts obtained from three different lines from each transgenic white and red radish plant were utilized to evaluate the efficiency of the expressed IFN-α2a protein as an antiviral and anticancer agent.

### 3.3. Antiviral Activity of the Recombinant IFN-α2a

The antiviral activity of recombinant IFN-α2a protein isolated from IFN-α2a transgenic white and red radish lines, total, and partially purified protein extracts was tested against VSV (Vesicular Stomatitis Virus) in two ways: post (direct) and pre-infection (indirect) of Vero cells with VSV. In this test, Peg-IFN and protein extracts from wild-type white and red radish were used as positive and negative controls, respectively. The results of this assay are shown in Table 1.

Protein extracts from transgenic white and red *Raphanus sativus* L. are clearly efficacious in inhibiting and decreasing VSV development and reproduction (Table 1). In the case of transgenic white radish protein extracts, it was observed that enriched protein fractions of IFN-wRs-2 transgenic plants (P-IFN-wRs in Table 1) have approximately 8 to and 27-fold more antiviral activity in direct and indirect measurements, respectively than wild-type white radish protein extracts.

Total protein extracts isolated from IFN-wRs-1, 2, and 3 showed approximately 6-fold and 22-fold more suppression of VSV activity compared to the corresponding wild-type protein extracts for direct and indirect measurements, respectively. Similar antiviral activity was observed for recombinant IFN-α2a protein isolated from transgenic red radish plants compared to transgenic white radish plants. Partially purified protein extracts isolated from IFN-rRs-2 transgenic plants (P-IFN-rRs in Table 1) showed 8.8-fold and 27.3-fold more antiviral activity compared to wild-type red radish protein extracts for the direct and indirect measurements, respectively. Total protein extracts isolated from IFN-rRs-1, 2, and 3 recorded an average of 6-fold and 24-fold more suppression of VSV activity compared to the corresponding wild-type protein extracts for direct and indirect assays, respectively. It is also obvious that under the assay conditions, recombinant IFN-α2a protein extracts from both transgenic radish genotypes have equivalent VSV antiviral activity to commercially available peg-IFN.

Previous research has demonstrated that recombinant IFN-α has antiviral and immune-regulatory properties, including the stimulation of class I histocompatibility complex antigens, which causes lymphocytes and macrophages to unleash cytotoxic activities [44]. Moreover, it has been shown that IFN-α acts either by the direct inhibition of cell growth or through the excitation and activation of the immune system [45]. Research implemented for the production of recombinant IFN-α in *Chlamydomonas reinhardtii* [39] and rice [46] or IFN-γ in soybean [17] showed similar inhibitory effects in suppressing the growth and reproduction of VSV.

When recombinant IFN-α2a derived from both types of transgenic radish plants is administered to Vero cells before viral infection, the overall results demonstrate that it functions better as an antiviral agent (indirect assay). As a result, it can be concluded that recombinant IFN-α2a produced in transgenic radish plants is functionally active as an antiviral drug, and its activity is mediated via the cellular immune system’s excitation and activation.

### 3.4. Antitumor Activity of the Recombinant IFN-α2a Isolated from Transgenic White and Red Raphanus sativus L. plants

#### 3.4.1. Antitumor Effect of the Recombinant IFN-α2a on Hep-G2 Tumor Cell Line (In Vitro Assay)

The research was expanded to assess the impact of recombinant IFN-α2a protein produced in transgenic white and red *Raphanus sativus* L. plants on the growth of the human Hep-G2 (human hepatocellular carcinoma) tumor cell line in vitro. The MTT test was used, as stated in the materials and methods section (Section 2.8). In this test, total and enriched fractions of recombinant IFN-α2a protein extracts were employed. In addition to the commercially available peg-IFN, protein extracts from wild-type white and red radish plants were employed as positive and negative controls, respectively. Partially purified recombinant IFN-α2a protein and protein extracts derived from three independent lines from each IFN-α2a transgenic white and red *Raphanus sativus* L. plant demonstrate a strong antiproliferative effect on the Hep-G2 tumor cell line (Table 2). In total protein extracts isolated from transgenic white and red radish plants, the average IC_50_ values for recombinant IFN-α2a are 8.8 μg/mLand 10.7 μg/mL, respectively. However, the IC_50_ values measured for wild-type white and red radish plants are ~531 μg/mL and ~468 μg/mL, respectively. Thus, the observed IC_50_ values for transgenic white and red radish protein extracts showed ~60-fold and ~43-fold more antitumor activity compared to wild-type protein extracts, respectively. Moreover, it is obvious that IC_50_ values recorded for recombinant IFN-α2a protein are comparable to IC_50_ values recorded for the commercially available Peg-IFN (see Table 2). This assay result demonstrates the functionality of the recombinant IFN-α2a produced in transgenic radish plants against Hep-G2 tumor cells.

Antitumor properties of plant-, microalgae-, and microorganism-derived protein extracts have been widely tested using Hep-G2 tumor cell lines. Many proteins were tested utilizing Hep-G2 tumor cell lines, including *Helicobacter pylori* CCUG 17874 [47], *E. coli* L-asparaginase (IC_50_: 46 μg/mL) [48], and *Penicillium brevicompactum* NRC 829 (IC_50_: 43.3 μg/mL) [49]. Furthermore, many kinds of human interferon, such as hIFN-α, hIFN-β, and hIFN-γ, have been widely researched for their potential to suppress the development and proliferation of Hep-G2 and Hep3B cells [39,50,51]. All of these reports supported the findings of the current investigation. Thus, IFN-α2a expressed in transgenic white and red radish plants has effective antitumor properties against Hep-G2 tumor cell lines.

#### 3.4.2. Effect of Recombinant IFN-α2a on Hep-G2 Cell Apoptosis

In addition to the CPE assay (Section 3.4.1), the antitumor properties of the recombinant IFN-α2a produced in transgenic white and red radish plants on Hep-G2 cell apoptosis were studied. This assay is based on the application of the IC_50_ concentration of the prepared recombinant IFN-α2a to growing Hep-G2 cells. In addition to wild-type (negative control) and Peg-IFN (positive control), partially purified IFN-α2a proteins and IFN-α2a total protein extracts isolated from transgenic white and red radish plants were employed for this purpose. In the same assay, untreated Hep-G2 cells were employed as an untreated control. Table 3 shows the results of this test. When P-IFN-wRs, IFN-wRs-1, IFN-wRs-2, and IFN-wRs-3 protein extracts are applied to Hep-G2 cells, the percentage of normal cells is greatly reduced (5.40%, 8.47%, 8.58%, and 7.82 %, respectively). This was accompanied by a significant increase in necrotic cell percentage (82.6%, 63.8%, 79.6%, and 57.1%, respectively) as shown in Table 3. A relatively and significantly lower effect was observed for transgenic red radish plants, 5.10%, 7.17%, 7.51%, and 6.82% (normal cells) and 81.2, 60.7, 76.2, and 54.1% (necrotic cells) were observed for P-IFN-rRs, IFN-rRs-1, IFN-rRs-2, and IFN-rRs-3, respectively. Negligible inhibitory effects were observed for white and red radish wild-type controls, the recorded values for WT protein extracts (~0.67% necrosis), and untreated Hep-G2 cells (0.59% necrosis). For all of the measured parameters, normal, early apoptosis, late apoptosis, and necrotic cells, it is evident that recombinant IFN-α2a isolated from white radish has relatively higher effects than the recombinant IFN-α2a isolated from transgenic red radish under the assay conditions.

IFN has been utilized in the treatment of hepatocellular carcinoma (HCC), albeit with some mixed results [33,52,53,54]. The current antitumor assay results, on the other hand, demonstrate that recombinant IFN-α2a generated in transgenic radish plants is capable of stopping cell growth and inducing cell apoptosis. Despite all of the research done to figure out how IFNs cause apoptosis, it is still unknown how a tumor cell chooses between cell death and growth arrest. The antitumor characteristics of recombinant IFN-α2a produced in transgenic white and red *Raphanus sativus* L. plants against Hep-G2 cell line assays, revealed in the current investigation, suggest recombinant IFN-α2a’s usefulness as an antitumor agent and support the use of radish plants as a promising bio-system for the production of functionally active human IFN-α2a.

## 4. Conclusions

Producing active biopharmaceuticals in *Raphanus sativus* L. plants has several advantages, including the ability to create proteins with glycosylation and/or post-translational modifications in addition to the absence of toxins and human pathogens. This last feature has the potential to reduce the number of purification steps required in downstream manufacturing. In addition, *Raphanus sativus* L. plants grow quickly and require little time from transformation to protein production. *Raphanus sativus* is a suitable system for the manufacture of pharmaceutically active human interferon because of these properties, as well as the findings of the current investigation. The generated IFN-α2a expressed in radish plants was proven to be functionally active as an anticancer and antiviral drug according to molecular and biochemical investigations. However, more research is needed, such as optimizing transgenic codon usage, finding strong constitutive promoters to boost gene expression, and optimizing both the production and administration of the produced proteins.

## Figures and Tables

**Figure 1 bioengineering-09-00381-f001:**
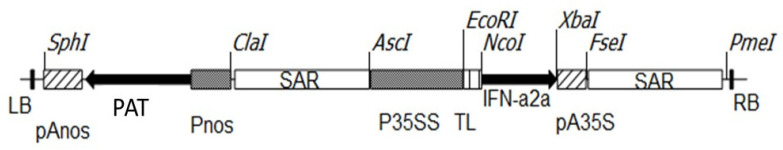
IFN-α2a gene expression construct.pTRA-PT, a derivative of pPAM (gi13508478) with a constitutive CaMV p35SS promoter and the 5′ UTR of the Tobacco Leader peptide (TL) was used for IFN-α2a expression in *Raphanus sativus* L. plants. RB/LB: right and left border sequences of nopaline-Ti-plasmid pTiT37. pAnos: polyadenylationsignal of nopaline synthetase gene derived from *Agrobacterium tumefaciens*. PAT: phosphinothricin acetyltransferase gene that confers resistance to BASTA (phosphinothricin). Pnos: nopaline synthase promoter of *A. tumefaciens*. IFN-α2a expression cassette is flanked by the 3′ UTR of CaMV (pA35S) and the scaffold attachment region (SAR) of the tobacco RB7 gene (gi3522871).

**Figure 2 bioengineering-09-00381-f002:**
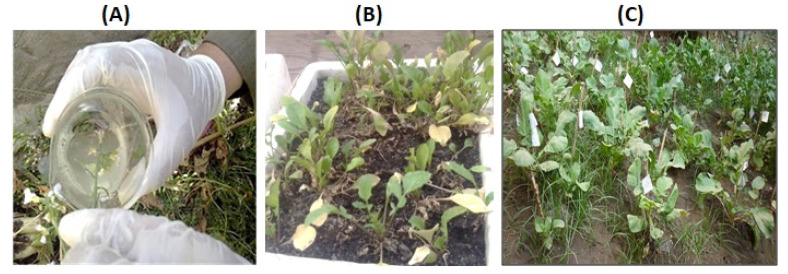
Representative photographs of *Raphanus sativus* L. plants. (**A**) Floral dip transformation of *Raphanus sativus* L. plants. (**B**) Selection of transgenic *Raphanus sativus* L. plants after spraying the foliar leaves with BASTA solution (phosphinothricin; 25 µg/mL) 6 times at 3-day intervals. (**C**) T_1_ plants of IFN-α2a transgenic *Raphanus Sativus* L. plants grown in soil.

**Figure 3 bioengineering-09-00381-f003:**
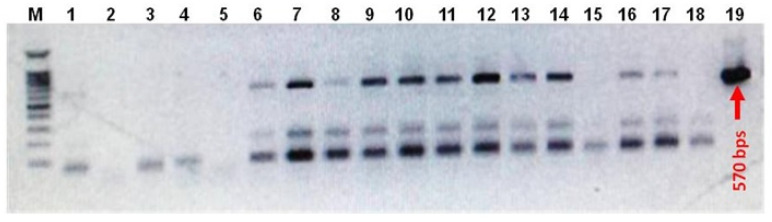
DNA gel electrophoresis of PCR screening for transgenic white and red *Raphanus sativus* L. plants. Shown is the ethidium bromide-mediated fluorescence of DNA fragments after UV excitation. Products of PCR screening of genomic DNA isolated from transgenic white and red *Raphanus sativus* L. plants were separated on a 1 % (*w*/*v*) agarose gel in 1x TAE for 50 min at 100 V. M: 100 bps DNA ladder, lane 1–12: PCR products of the screened white *Raphanus sativus L* plants, lane 13–17: PCR products of the screened red *Raphanus sativus* L. plants, lane 18: negative PCR water control, and lane 19: pTRA-PT-IFN-α2a plasmid positive control.

**Figure 4 bioengineering-09-00381-f004:**
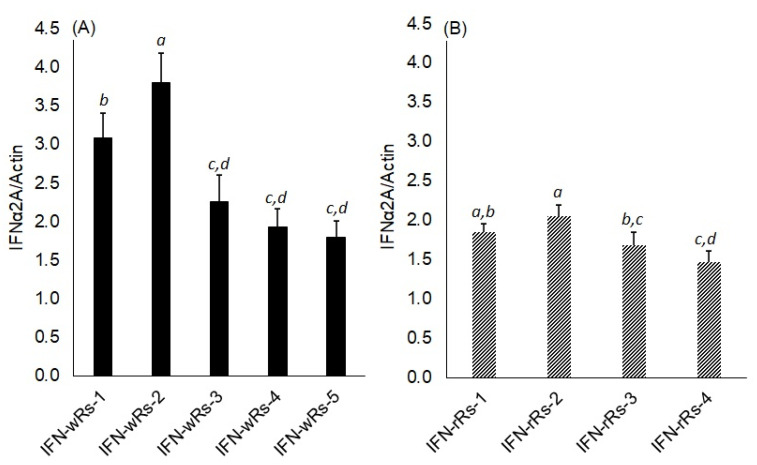
RT-PCR analysis of IFNα2a gene expression in transgenic white (**A**) and red (**B**) *Raphanus sativus* L. plants. IFN-α2a mRNA transcripts was measured by real-time qRT-PCR and calculated in arbitrary units compared to a standard dilution series. IFN-wRs 1-5: samples from five independent IFNα2a transgenic white *Raphanus sativus* L. plants, IFN-rRs 1-4: samples from four independent IFNα2a transgenic red *Raphanus sativus* L. plants. Data are represented as the average of at least three independent measurements of RNA preparations ± SE. Different small letters represent significant differences; *p* = 0.05.

**Figure 5 bioengineering-09-00381-f005:**
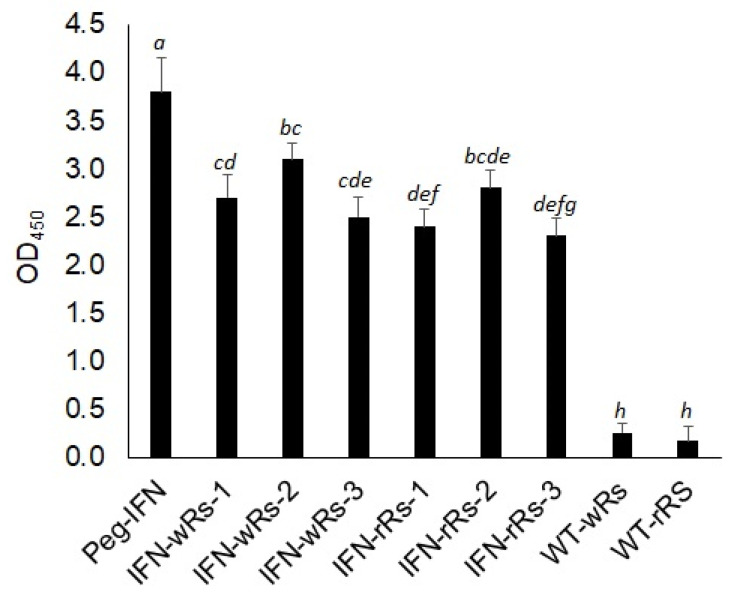
Enzyme-linked Immunosorbent Assay (ELISA) detection for IFN-α2a in wild-type and transgenic *Raphanus sativus* L. plants. Shown are OD_450_ values recorded for recombinant IFN-α2a samples and Peg-IFN (commercially available IFN-α (Pegasys^®^; Hoffmann-LaRoche, Basel, Switzerland)). IFN-wRs-1–3: total protein extracts isolated from three independent transgenic lines of IFN-α2a- transgenic white *Raphanus sativus* L. plants. IFN-rRs-1–3: total protein extracts isolated from three independent transgenic lines of IFN-α2a-transgenic red *Raphanus sativus* L., WT-wRs: wild-type white *Raphanus sativus* L. protein extracts. WT-rRs: wild-type red *Raphanus sativus* L. protein extracts. Data represent the average OD_450_ values recorded from at least 4 independent measurements ± SE. Different small letters represent significant differences; *p* = 0.01.

**Table 1 bioengineering-09-00381-t001:** The recombinant IFN-α2a protein antiviral activity.

Sample Code	Reduction in VSV Titer (%)	
	Direct	Indirect
Peg-IFN	7.8 ± 0.25	75.4 ± 6.1
P-IFN-wRs	6.7 ± 0.15	63.2 ± 3.2
IFN-wRs-1	4.8 ± 0.20	50.8 ± 1.2
IFN-wRs-2	5.5 ± 0.14	52.3 ± 1.6
IFN-wRs-3	4.1 ± 0.21	48.2 ± 1.3
P-IFN-rRs	6.2 ± 0.18	60.2 ± 2.2
IFN-rRs-1	4.4 ± 0.16	53.4 ± 1.2
IFN-rRs-2	4.8 ± 0.11	55.6 ± 2.1
IFN-rRs-3	3.8 ± 0.17	49. 8 ± 1.8
WT-wRS	0.8 ± 0.06	02.3 ± 0.1
WT-rRS	0.7 ± 0.05	02.0 ± 0.2

Peg-IFN: Commercially available IFN-α2a (Pegasys^®^; Hoffmann-LaRoche, Basel, Switzerland), P-IFN-wRs: partially purified IFN protein from IFN-α2a transgenic white radish, IFN-wRs-1–3: total protein extracts isolated from three independent IFN-α2a transgenic lines of white radish, P-IFN-rRs: partially purified IFN protein from IFN-α2a-transgenic red radish, IFN-rRs-1–3: total protein extracts isolated from three independent IFN-α2a transgenic lines of red radish, WT-wRs: wild-type white radish protein extracts, WT-rRs: wild-type red radish protein extracts. Data represents the average of the percentage of reduction in VSV titer post-treatment of at least 6 independent measurements ± SE. *t-*test; *p* ˂ 0.001.

**Table 2 bioengineering-09-00381-t002:** Cytotoxicity effect of recombinant INFα2a against Hep-G2 cell line.

Sample Name	IC_50_ (μgProtein)	SE
Peg-IFN	6.2	0.62
P-IFN-wRs	7.4	0.77
IFN-wRs-1	8.9	0.47
IFN-wRs-2	8.5	0.55
IFN-wRs-3	9.2	0.81
P-IFN-rRs	8.2	0.93
IFN-rRs-1	10.3	0.65
IFN-rRs-2	10.1	0.53
IFN-rRs-3	11.7	0.68
WT-wRS	531.1	3.62
WT-rRS	468.5	4.23

P-IFN: commercially available IFN-α2a (Pegasys^®^; Hoffmann-LaRoche, Basel, Switzerland), P-IFN-wRs: partially purified IFN protein from IFN-α2a-transgenic white radish, IFN-wRs-1–3: total protein extracts from three independent transgenic lines of IFN-α2a-transgenic white radish, P-IFN-rRs: partially purified IFN protein from IFN-α2a-transgenic red radish, IFN-rRs-1–3: total protein extracts from three independent transgenic lines of IFN-α2a-transgenic red radish, WT-wRS: wild-type white radish protein extracts, WT-rRS: wild-type red radish protein extracts. Data represents the average of the IC_50_ values recorded from at least 3 independent measurements ± SE. *t-*test; *p*-value ˂ 0.001.

**Table 3 bioengineering-09-00381-t003:** Effect of recombinant IFN-α2a on Hep-G2 cell apoptosis at various stages.

Sample Name	Normal Cells (%)	Early Apoptosis (%)	Late Apoptosis (%)	Necrotic Cells (%)
Control	99.4	0.03	0.07	0.59
Peg-IFN	4.70	1.02	54.55	87.8
P-IFN-wRs	5.40	0.14	37.42	82.6
IFN-wRs-1	8.47	0.07	26.4	63.8
IFN-wRs-2	8.58	0.08	28.7	79.6
IFN-wRs-3	7.82	0.06	25.2	57.1
P-IFN-rRs	5.10	0.12	39.52	81.2
IFN-rRs-1	7.17	0.12	24.3	60.7
IFN-rRs-2	7.51	0.18	26.7	76.2
IFN-rRs-3	6.82	0.16	23.4	54.1
WT-wRS	96.7	0.72	2.21	0.62
WT-rRS	95.9	0.83	2.11	0.72

Control: untreated Hep-G2 cells; *t-*test; *p*-value ˂ 0.001.

## Data Availability

Not applicable.

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
