# Peer review of "A Biotechnological Approach for the Production of Pharmaceutically Active Human Interferon-α from *Raphanus sativus* L. Plants"

_bioengineering, 2022, doi:10.3390/bioengineering9080381_

Round 1

Reviewer 2 Report

In this manuscript, the authors characterize and test human IFNa activity in extracts from red and white radishes. The manuscript is well written and thorough although there are a few issues for consideration.

1. In Figure 4, why are there any bands in the H2O control?

2. In the RNA analysis, how did the authors rule out DNA contamination, given the IFN gene has no introns?

3. Have the authors mixed control recombinant human IFN with WT red or white radish extracts to see of the extracts enhance or depress the relative IFN activity.

4. Have the authors evaluated the IFN extracts for their ability to induce phospho-STAT 1 and/or p-STAT2?

5. Although certainly not expected, have the extracts been tested for endotoxin?

6. Although beyond the scope of this study, is  the plant IFNa glycosylated (see abstract below) ?

Biochem J. 1991 Jun 1; 276(Pt 2): 511–518. doi: 10.1042/bj2760511 PMCID: PMC1151121 PMID: 2049076

Natural human interferon-alpha 2 is O-glycosylated.

G R Adolf, I Kalsner, H Ahorn, I Maurer-Fogy, and K Cantell Author information Copyright and License information Disclaimer   This article has been cited by other articles in PMC.

Abstract

Natural human interferon alpha 2 (IFN-alpha 2) was isolated from a preparation of partially purified human leucocyte IFN by monoclonal-antibody immunoaffinity chromatography. The purified protein had a specific activity of 1.5 x 10(8) i.u./mg; it was estimated to constitute 10-20% of the total antiviral activity of leucocyte IFN. N-Terminal amino-acid-sequence analysis identified the subspecies IFN-alpha 2b and/or IFN-alpha 2c, whereas IFN-alpha 2a was not detectable. The structure of natural IFN-alpha 2 was found to differ from that of its recombinant (Escherichia coli-derived) equivalent. First, reverse-phase h.p.l.c. showed that natural IFN-alpha 2 was significantly more hydrophilic then expected. Secondly, the apparent molecular mass of the natural protein determined by SDS/PAGE was higher than that of recombinant IFN-alpha 2; incubation under mild alkaline conditions known to eliminate O-linked carbohydrates resulted in a reduction of the apparent molecular mass to that of the recombinant protein. On sequence analysis of proteolytic peptides, Thr-106 was found to be modified. These results suggested that Thr-106 of natural IFN-alpha 2 carries O-linked carbohydrates. Reverse-phase h.p.l.c. as well as SDS/PAGE of natural IFN-alpha 2 showed that glycosylation is heterogeneous. For characterization of the carbohydrate moieties, the protein was treated with neuraminidase and/or O-glycanase and analysed by gel electrophoresis; in addition, glycopeptides obtained by proteinase digestion and separated by h.p.l.c. were characterized by sequence analysis and m.s. Further information on the composition of the glycans was obtained by monosaccharide analysis. The results indicate that natural IFN-alpha 2 contains the disaccharide galactosyl-N-acetylgalactosamine (Gal-GalNAc) linked to Thr-106. In part of the molecules, this core carbohydrate carries (alpha-)N-acetylneuraminic acid, whereas a disaccharide, probably N-acetyl-lactosamine, is bound to Gal-GalNAc in another proportion of the protein. Further glycosylation isomers are present in small amounts. As IFN-alpha 2 is the only IFN-alpha species with a threonine residue at position 106, it may represent the only O-glycosylated human IFN-alpha protein.

Reviewer 3 Report

I reviewed the manuscript entitled, A biotechnological approach for production of pharmaceutically active human interferon-α from Raphanus sativus L plants. The manuscript is well written with appropriate introduction and methodology. Tables and Figures are appropriate. Conclusions reflect the findings. Based on these observations, I recommend this manuscript for publication consideration after addressing below suggestions.

Line 3. Scientific name must be in Italics

Abstract: this section should be revised with clear objectives of the research

Although the introduction is written very well, the objectives of the research were not clear. Please revise accordingly.

1.1. Plant material: who identified the plant and its number should be mentioned.

Line 123: 3 weeks... sentence should not start with number.

Line 214: “C” should be capital letter

Line 232 and 234 238: (Sigma Aldrich, Germany). ….. write the location of company  

Methodology is well written

Results and discussion

Figure 4. please perform the statistical analysis (for both A and B)

Figure 5. please perform the statistical analysis

Table 1. please perform the statistical analysis

References are not according to the journal format. For example, scientific names must be in Italics, see ref 2, 12, and others. The format is inconsistent

Author Response

This manuscript is a resubmission of an earlier submission. The following is a list of the peer review reports and author responses from that submission.